# In the Eye of MLLM: Benchmarking Egocentric Video Intent Understanding with Gaze-Guided Prompting

**Taiying Peng**[1] **Jiacheng Hua**[2] **Miao Liu**[2†] **Feng Lu**[1†]

[1]State Key Laboratory of VR Technology and Systems, School of CSE, Beihang University

[2]College of AI, Tsinghua University

## Abstract

The emergence of advanced multimodal large language models (MLLMs) has significantly enhanced AI assistants' ability to process complex information across modalities. Recently, egocentric videos, by directly capturing user focus, actions, and context in an unified coordinate, offer an exciting opportunity to enable proactive and personalized AI user experiences with MLLMs. However, existing benchmarks overlook the crucial role of gaze as an indicator of user intent. To address this gap, we introduce EgoGazeVQA, an egocentric gaze-guided video question answering benchmark that leverages gaze information to improve the understanding of longer daily-life videos. EgoGazeVQA consists of gaze-based QA pairs generated by MLLMs and refined by human annotators. Our experiments reveal that existing MLLMs struggle to accurately interpret user intentions. In contrast, our gaze-guided intent prompting methods significantly enhance performance by integrating spatial, temporal, and intent-related cues. We further conduct experiments on gaze-related fine-tuning and analyze how gaze estimation accuracy impacts prompting effectiveness. These results underscore the value of gaze for more personalized and effective AI assistants in egocentric settings. Project page: https://taiyi98.github.io/projects/EgoGazeVQA.

## 1 Introduction

The rapid advancement of multimodal large language models (MLLMs) [16, 31, 5, 32] has enabled highly intelligent AI assistants capable of understanding and generating complex information across modalities. This capability is not only valuable for enhancing current AI systems but also crucial for advancing future human–AI collaboration systems, thereby enabling new AI paradigms such as cobodied AI [27]. However, to effectively collaborate with users, existing systems often require them to design intricate prompts to convey their intentions and personalize the assistant's responses. This challenge raises an essential question: Can existing MLLM enable a more proactive AI user experience– one where the assistant seamlessly analyzes the users' underlying intention and delivers personalized content without heavily replying on complex prompting?

Egocentric videos may serve as an ideal solution to this challenge: the unique viewpoint aligns human visual perception, attention, action, and surrounding scene-context within the same egocentric coordinate system, and thereby offers valuable insights into an individual's experiences, actions, and intentions. Despite the potential of egocentric capture to pave the way for a more personalized and proactive AI user experience [17], existing MLLMs have been predominantly developed and

---

† Corresponding authors.

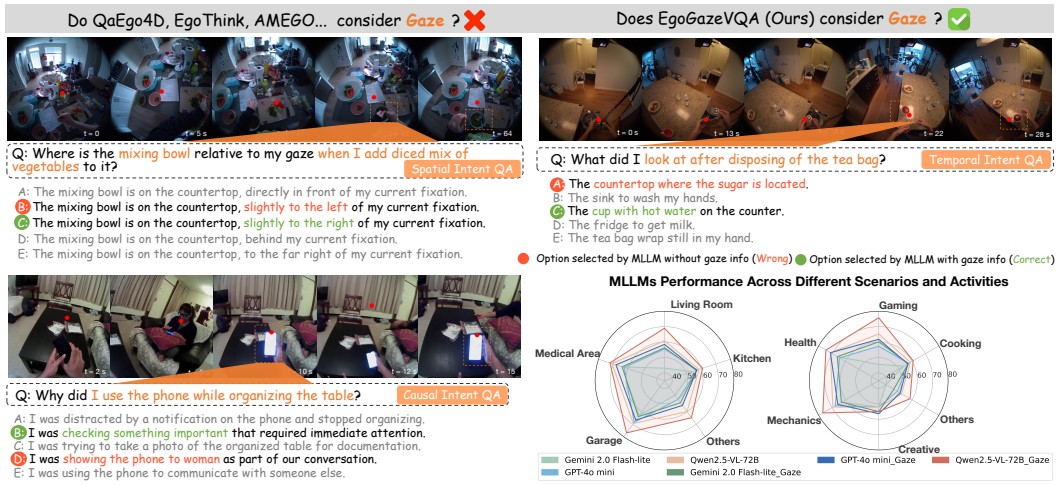

Figure 1: We propose EgoGazeVQA, the first MLLM benchmark that incorporates essential gaze signals for understanding user intent in egocentric settings. We present examples of Spatial, Temporal, and Causal Intent QA, demonstrating how gaze information improves MLLMs' performance. Correct and incorrect predictions by MLLMs with and without gaze cues are highlighted. Radar charts compare model performance across different scenarios and activities, showing consistent gains with our gaze-guide prompting strategy.

evaluated using exocentric video benchmarks [37, 39, 22, 9, 33, 23, 7, 34]. Recent works introduced several egocentric VQA benchmarks for Episodic Memory [3], Long-Form Video reasoning [30], Scene Layout analysis [15], and Scene-Text understanding [39]. However, these benchmarks have overlooked a crucial egocentric cue—*Gaze*—which provides direct insights into user attention [2] and intent. For egocentric AI assistant, the majority of user questions are likely to be inherently conditioned on what the user is looking at. Taking the examples in Figure 1, the MLLMs may run into failure mode when addressing questions about user fixation without gaze signals. To bridge this gap, we introduce a timely benchmark – EgoGazeVQA, an Egocentric Gaze-Guided Video Question Answering Benchmark, specifically designed to assess whether MLLMs can utilize egocentric gaze signals for analyzing daily-routine videos.

The EgoGazeVQA dataset is constructed by processing egocentric video clips to extract descriptive captions for each frame and gaze tracking data, capturing the user's focus through (x, y) coordinates. We feed this information into a strong MLLM model to generate spatial/temporal-aware and intent-related questions along with multiple plausible answer options. The generated Q&A pairs are reviewed by human annotators for relevance, answerability, fluency, accuracy, conciseness, and difficulty, ensuring high-quality and contextually rich data that reflects real-world egocentric interactions.

We conduct comprehensive experiments on the proposed benchmark and observe that existing MLLMs often struggle to accurately interpret user intentions from egocentric videos alone. This limitation stems from the prevalent use of global image frames to construct visual tokens—an approach that provides broad context but fails to capture the camera wearer's explicit signals of intent. To address this issue, we explore several Gaze-Guided Prompting Strategies aimed at enhancing the model's ability to understand user focus and intention. Moreover, since gaze-related signals are rarely seen during MLLM training, these models exhibit limited capacity to interpret gaze cues. To mitigate this, we further investigate how LoRA-based fine-tuning can help bridge this gap.

To summarize, we make the following contributions:

- We introduce EgoGazeVQA, the first egocentric gaze-guided video intent QA benchmark designed to evaluate whether MLLMs can leverage gaze information to enhance the understanding of human intentions in daily-life videos.

- We introduce three gaze-guided prompting strategies to improve MLLM's ability to interpret spatial, temporal, and intent-related cues.

- We present empirical analysis demonstrating how fine-tuning can enhances a model's ability to leverage gaze signals for understanding human intentions from an egocentric perspective.

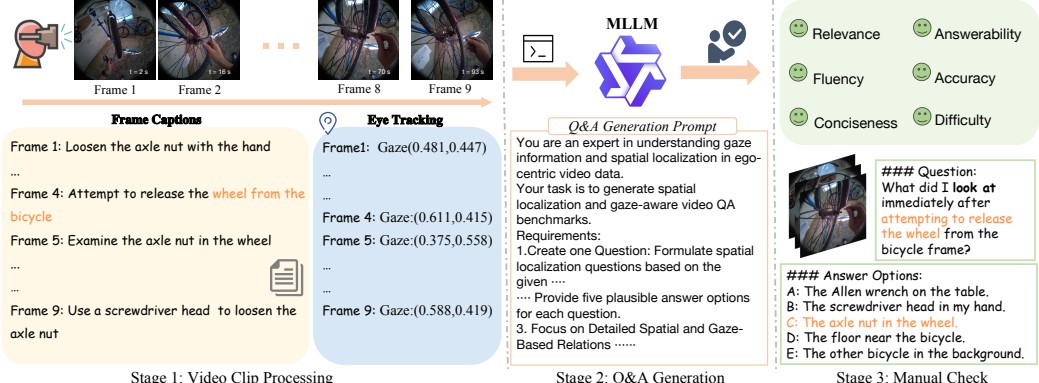

Figure 2: Construction pipeline of the EgoGazeVQA. We craft the EgoGazeVQA in three steps. Stage 1: Egocentric video clips are processed to extract frame captions and gaze coordinates to capture user focus. Stage 2: A MLLM model generates spatial/temporal-ware and intention-related Q&A pairs using a customized prompt. Stage 3: Human annotators manually review the generated Q&A pairs for several important quality dimensions to ensure high-quality data.

## 2 Related Work

**Video QA benchmarks.** Existing Video QA benchmarks primarily focus on exocentric video understanding [37, 39, 22, 9, 33, 23, 7, 34]. For instance, NExT-QA [39] focuses on temporal, causal, and descriptive questions for short clips, while IntentQA [22] targets reasoning about human intent in longer videos. Fu *et al.* [9] introduced VideoMME, a diverse video analysis benchmark. Lvbenchmark [33] was developed to evaluate MLLM's capabilities in understanding extremely long video. Our work is more closely connected to recent efforts on egocentric VQA benchmark. QaEgo4D [3] focuses on episodic memory-based, while EgoSchema [28] addresses reasoning over long-form egocentric videos. Additionally, Huang *et al.* [15] proposed the VideoMindPalace Benchmark to evaluate MLLM's ability to reason about spatial-temporal and 3D scene layouts. Zhou *et al.* [39] developed a scene-text QA benchmark using egocentric data. In contrast to previous efforts, we introduce EgoGazeVQA—the first benchmark that integrates user gaze data to enable Egocentric Video QA focused on understanding personal intent, actions, and contextual information.

**Egocentric gaze and actions.** The connection between gaze and actions in the egocentric coordinate has been studied in a few recent works. Early efforts leverage gaze-indexed visual features for egocentric action recognition [26]. Shen *et al.* [30] proposed using the gaze event toward objects for egocentric action forecasting. Li *et al.*developed a novel method for the joint learning of egocentric gaze and actions [25, 24]. Joint modeling between egocentric gaze and actions with CNN has also been studied in [13]. Huang *et al.* [14] introduced a novel model to learn temporal attention transitions from video features that reflect drastic gaze movements. More recently, Lai *et al.* [20, 19, 18] redesigned transformer architectures for gaze estimation and anticipation. Despite significant progress in understanding egocentric gaze and actions, previous work never explored how gaze cues can assist MLLM answer questions about user intentions and actions from an egocentric perspective. Recently, Konrad *et al.* [17] introduced GazeGPT to showcase gaze data can help design a better MLLM UI, yet a standardized benchmark and evaluation is missing from this study. Our work seeks to bridge this gap by introducing the Video Egocentric VQA benchmark, which examines models capability on using egocentric gaze cues to infer user intentions and personal context.

## 3 EgoGazeVQA

### 3.1 Benchmark construction

Our video data was sourced from three major egocentric video datasets with eye tracking data: Ego4D [11], EgoExo4D [12], EGTEA Gaze+ [25] and integrated. We leveraged existing multimodal large models to generate initial video–question–answer pairs, ensuring diversity in both spatiotemporal context and causal logic. Each of the generated QA items went thorough human verification, to ensure high quality and consistency in the final benchmark.

Table 1: Distribution of EgoGazeVQA categories and related question-answer pairs.

| Category | QA Pairs | Q&A Examples |
|---|---|---|
| *EgoGazeVQA-Scenario* | | |
| Living Room | 371 | *Q:Why did I shuffle the cards while playing*
A: I was trying to ensure a fair game by mixing the cards thoroughly. |
| Kitchen | 1024 | *Q:What did I look at immediately after reading the recipe for the first time?*
A:The toasted sesame oil on the counter top. |
| Medical Area | 72 | *Q:Why did I interlock my hands on the center of the patient's chest?*
A: I was ensuring proper hand placement for effective chest compressions. |
| Garage | 108 | *Q: Why did I use an Allen wrench to loosen the axle nut after initially trying with my hand and a regular wrench ?*
A: I was distracted by the nearby bicycle and kept adjusting it instead. |
| Others | 182 | *Q: Where is the cat relative to my gaze direction when I hold it?*
A: The cat is on the carpet, slightly below and to the right of my current fixation. |
| *EgoGazeVQA-Activity* | | |
| Creative | 69 | *Q:What did I look at after getting the soy sauce from the countertop?*
A: The recipe manual. |
| Gaming | 314 | *Q:What did the viewer look at immediately after picking up the game controller?*
A: The television screen displaying the game instructions. |
| Cooking | 1019 | *Q: Where is the skillet relative to my gaze when I pour the egg mixture into it?*
A: The skillet is on the stove, slightly to the right of my current fixation. |
| Mechanics | 108 | *Q:What did I look at after pulling the tire and the inner tube out of the rim?*
A: The inner tube for any damage or splits. |
| Health | 59 | *Q: What did I look at immediately after tapping the patient to check their response?*
A: The patient's face |
| Others | 188 | *Q: Why did I look at man X while lady Z was playing*
A: I was checking if man X had any cards to trade with lady Z. |

**Video clip preprocessing.** The sourced egocentric videos cover everyday scenarios, ranging from indoor and outdoor social environments to health-related, mechanical, and culinary activities. Ego4D contains 3,600 hours (and counting) of densely narrated videos spanning benchmarks such as Episodic Memory, Hands and Objects, and Forecasting. From Ego4D, we sampled 31 hours of videos with eye tracking data, and further segmented into 168 video clips (averaging from 30 seconds to 1 minute in duration) based on the narration labels to ensure rich spatial-temporal information. EgoExo4D, which includes 1,286.30 hours of video, also provides multi-channel sensory data such as 7-channel audio, IMU, eyegaze, two grayscale SLAM cameras, and 3D environment point clouds. We focused on the videos within the KeyStep tasks that feature essential text descriptions and gaze data, ultimately forming 263 annotated clips after applying the same segmentation criteria. Finally, EGTEA Gaze+ offers 28 hours (de-identified) of cooking activities across 86 sessions with 32 subjects, accompanied by 30Hz gaze tracking and audio. Following similar sampling protocols, we obtained over 300 video-text-gaze annotation files. This approach ensures robust coverage of diverse spatiotemporal contexts, enriched by explicit gaze tracking annotations.

**QA pairs generation via advanced MLLMs.** Manually creating video–question–answer data from gaze information and textual descriptions is a labor-intensive process and risks leading to limited question variety. To address this, we employ advanced fully-featured multimodal large language models (MLLMs), such as Qwen2.5-VL, following a structured procedure. We provide a visual illustration of our method in Figure 2. First, we extract video frames with meaningful descriptors to ensure semantic alignment between visual content and textual annotations. We then group every nine frames into a keyframe-based segment, each paired with normalized gaze coordinates captured at those frames. Next, we feed these nine frames and their gaze coordinates as prompts into Qwen2.5-VL to generate three sets of QA pairs. Each QA pair contains five multiple-choice options, with exactly one correct answer. Given that Ego4D and EgoExo4D have detailed descriptions, each video can yield up to three QA sets. However, for EGTEA Gaze+, where descriptions are sparser, a single QA set is generated per video to maintain data diversity. The prompts are designed to: (1) fully exploit the first-person viewpoint and gaze focus, capturing realistic user–object interactions; (2) combine gaze trajectories and event details to reflect user intentions and activity progression; and (3) include a range of distractor strategies—such as reverse-causal options, proximity traps, irrelevant yet visually

salient elements, and social-influence distractions—to ensure high-quality, diverse answer choices. The detailed prompt used in our pipeline can be found in the Appendix E.

**Human verification.** Human annotators evaluate each pair based on six key criteria: *relevance* (alignment with video content and gaze data), *answerability* (whether the questions can be answered using the provided information), *fluency* (linguistic naturalness and clarity), *accuracy* (correctness of the answers), *conciseness* (elimination of unnecessary details), and *difficulty* (maintaining a balanced range of question complexities). To ensure that the Q&A pairs are contextually rich, accurate, and reflective of real-world egocentric interactions, we verified the data and removed the samples that failed our quality criteria, yielding the final benchmark. The three most common types of errors identified during human review are summarized in Table 6

## 3.2 Benchmark characteristics

**Dataset statistics.** EgoGazeVQA is designed to strengthen users' comprehension of daily scenarios and activities, enabling more accurate reasoning about object interactions and user intentions. To better characterize the diverse contexts in this benchmark, we categorize the questions in two ways: (1) Scenario featuring: Living Room, Kitchen, Medical Area, Garage and Others; and (2) Activity type: Creative, Gaming, Cooking, Mechanics, Health and Others. Detailed distributions of videos and corresponding questions are presented in Table 1.

**Dataset comparison.** In Table 2, we compare EgoGazeVQA with several related benchmarks. Notably, EgoGazeVQA is the first to leverage eye gaze information to drive a model's understanding of user intentions and environmental awareness in daily scenarios, resulting in a collection of 1,700 QA pairs drawn from 900 videos. Additionally, we integrate user eye-tracking data to precisely record gaze positions in chronological order. Our

Table 2: Comparison of related egocentric VQA benchmarks. OE: Denotes open-ended QA. MC: Multiple Choice.

| Benchmark | Videos | Questions | Production | Gaze | Task |
|---|---|---|---|---|---|
| QaEgo4D [3] | 166 | 1854 | Human | ✗ | OE |
| EgoThink [6] | 700 | 700 | Auto | ✗ | OE |
| EgoTextVQA [39] | 1507 | 7064 | Auto | ✗ | OE |
| EgoSchema [28] | 5063 | 5063 | Auto | ✗ | MC |
| EgoMemoria [36] | 629 | 7026 | Auto | ✗ | MC |
| AMEGO [10] | 100 | 20500 | Auto | ✗ | MC |
| VideoMindPalace [15] | 200 | 1800 | Auto | ✗ | MC |
| EgoGazeVQA (Ours) | 913 | 1757 | Auto | ✓ | MC |

multiple-choice answers also include reverse-causal, spatial-proximity traps, social-influence traps, and high-salience distractors—all of which demand attention to the user's gaze for correct reasoning. This combination underscores EgoGazeVQA's unique focus on first-person perspective and intent-driven contextual understanding.

# 4 Experiments

## 4.1 Experimental setup

**Evaluation metric.** We assess the performance of the latest open-source and closed-source MLLMs on EgoGazeVQA from three dimensions–spatial, temporal, and causal—as illustrated in Figure 1. For our experiments, we use two closed-source API-based models—GPT-4o mini [29] and Gemini 2.0 Flash-Lite [8]—along with five state-of-the-art open-source MLLMs (InternVL2.5 [4], LLaVA-NeXT-Video [21], MiniCPM-o 2.6 [35], Qwen2.5-VL-72B [1] and Qwen2.5-VL-7B [1]). We provide each model with identical prompts to ensure fairness. During inference, we feed the multi-choice questions together with the corresponding list of video frames into the models, compare their selected answers to the ground-truth labels, and compute the resulting accuracy scores. Each multi-choice question has 5 options, therefore the by-chance accuracy is 20%.

## 4.2 Gaze-guided intent prompting

To investigate whether incorporating 2D gaze coordinates enhances a model's understanding of human intention on EgoGazeVQA benchmark, we explore three prompting strategies, as illustrated in Figure 3. Each strategy varies in how gaze information is encoded and presented to the MLLMs,

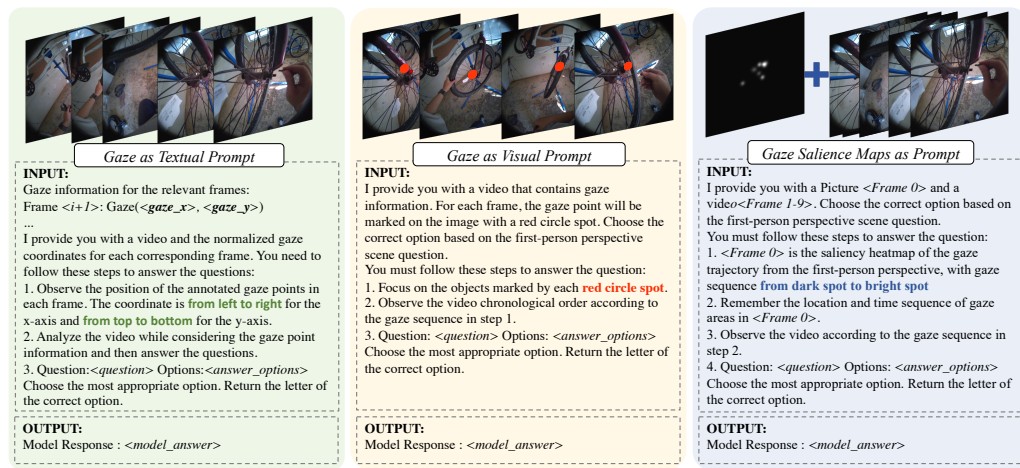

Figure 3: Gaze-guided prompting strategies in EgoGazeVQA. We experiment three gaze-guided prompting strategies on the EgoGazeVQA benchmark: Gaze as Textual Prompt (left), where gaze coordinates are presented as text inputs to guide model responses; Gaze as Visual Prompt (center), which highlights gaze points directly on video frames to inform answer selection; and Gaze Salience Maps as Prompt (right), utilizing heatmaps of gaze trajectories to provide contextual cues for understanding spatial and temporal intent. We evaluate demonstrate each strategy to enhance the MLLM's ability to interpret user focus and intent more accurately in Table 3.

allowing us to examine different levels of visual and textual alignment. The detailed inference prompt template can be found in Appendix E.

**Gaze as textual prompt (GazeT).** Because most modern MLLMs are built upon high-performing large language models, we encode frame-wise gaze information as two-dimensional, normalized coordinates and concatenate them directly with the prompt. This approach abstracts away potential size mismatches across videos, as all gaze data is standardized to a 0–1 range.

**Gaze as visual prompt (GazeV).** To fully leverage gaze as an attention-like mechanism in the visual domain, we mark each frame with a 25-pixel-radius red circle at the gaze coordinates. We then inform the MLLMs within the prompt that "the red circles represent highly attended regions." This visually driven approach more closely mirrors human viewing patterns by guiding the model to areas of heightened salience within the frame.

**Sequential gaze salience map as prompt (GazeS).** Considering the strong temporal correlation between the video context and gaze trajectory, we further model gaze over time using salience maps. Specifically, we progressively enhance the intensities of gaze saliency maps as the video advances and consolidate them into a single saliency

Table 3: Performance of prevailing MLLMs on EgoGazeVQA and effects of gaze-guided prompting. **T**: Gaze as Textual prompt. **V**: Gaze as Visual image prompt. **S**: sequential Gaze Salience map as prompt. The best accuracy results in each MLLMs are **bolded** respectively

| Method | Strategy | Spatial | Temporal | Causal | Avg. |
|---|---|---|---|---|---|
| Human | / | 80.7 | 75.6 | 95.2 | 83.8 |
| GPT-4o mini [29] | w/o | 48.4 | 47.1 | 75.6 | 57.0 |
| | T | 48.0 | **50.6** | 77.7 | **58.8** |
| | V | 46.8 | 49.7 | **78.9** | 58.5 |
| | S | **51.1** | 47.6 | 77.3 | 58.7 |
| Gemini 2.0 Flash-Lite [8] | w/o | 43.9 | 38.8 | 70.4 | 51.0 |
| | T | 45.6 | 44.0 | 77.7 | 55.8 |
| | V | 43.8 | 44.0 | 78.9 | 55.6 |
| | S | **45.7** | **45.6** | **80.3** | **57.2** |
| LLaVA-NeXT-Video-7B [21] | w/o | 44.9 | 45.3 | 74.5 | 54.9 |
| | T | 45.5 | 46.2 | 77.7 | 56.5 |
| | V | 46.1 | **46.6** | **79.7** | 57.5 |
| | S | **49.4** | 46.0 | 79.5 | **58.3** |
| MiniCPM-o 2.6-8B [31] | w/o | 40.8 | 34.5 | 32.5 | 35.9 |
| | T | **43.9** | 41.6 | 64.4 | 50.0 |
| | V | 41.5 | 41.6 | 67.3 | 50.2 |
| | S | 43.2 | **43.5** | **74.4** | **53.7** |
| InternVL2.5-8B [5] | w/o | 50.4 | 51.1 | 73.3 | 58.3 |
| | T | 51.8 | **52.7** | 75.7 | 60.1 |
| | V | **55.0** | 50.6 | **76.2** | **60.6** |
| | S | 53.2 | 51.0 | 75.7 | 59.9 |
| Qwen2.5-VL-7B [1] | w/o | 36.1 | 38.5 | 74.5 | 49.7 |
| | T | 36.9 | 35.3 | 76.2 | 49.5 |
| | V | 36.8 | 37.3 | 74.9 | 49.7 |
| | S | **41.9** | **40.4** | **77.9** | **53.4** |
| Qwen2.5-VL-72B [1] | w/o | 57.1 | 45.2 | 79.3 | 60.5 |
| | T | 60.0 | **50.7** | 84.2 | 65.0 |
| | V | 59.7 | 48.1 | 84.1 | 63.9 |
| | S | **64.3** | 50.3 | **84.3** | **66.3** |

map that captures both spatial and temporal intentions. Regions revisited by the user's gaze are assigned higher intensities, effectively mimicking human visual scanning patterns over time. Pseudocode for generating these salience maps is provided in Algorithm 1 of Appendix B.

## 4.3 Baseline Comparison with CLIP, EgoVLP, and BLIP-2

In Table 3, we report performance of representative MLLMs on EgoGazeVQA to establish demonstrative baselines and highlight the significant room for improvement on gaze-relevant tasks. In Table 4, we provide results using CLIP (ViT-H-14) and EgoVLP as simple baselines, where we compute the cosine similarity between the visual embeddings of the input and the text embeddings of each answer option, selecting the one with the highest similarity as the prediction. We also evaluate BLIP-2, which generates answers based on the visual embeddings and input question. We calculate the cross-entropy loss between the generated answer and each of the candidate options, and select the one with the lowest loss as the most likely match.

For both baselines, we follow the same video preprocessing pipeline described in the main paper. To mimic human visual attention, we use gaze fixation as an attentional map to select around 20% of each frame, resulting in what we term the *GazeMap* variant .

Table 4: Performance comparison of simple baseline models and Qwen2.5-VL-72B variants (i.e., CLIP-GazeMap, EgoVLP-GazeMap, and BLIP-2-GazeMap) on EgoGazeVQA.

| Method | Similarity | Performance |
|---|---|---|
| CLIP ViT-H-14 | Video-A | 21.63 |
| CLIP-GazeMap | Video-A | 21.79 |
| EgoVLP | Video-A | 22.95 |
| EgoVLP-GazeMap | Video-A | 22.65 |
| BLIP-2 | N/A | 27.60 |
| BLIP-2-GazeMap | N/A | 27.21 |
| Qwen2.5-VL-72B | N/A | 60.5 |
| Qwen2.5-VL-72B-GazeVisual | N/A | 63.9 |

Notably, these simple baselines perform only marginally better than random guessing and fall far short of human expert performance, indicating that CLIP-based methods struggle to tackle this challenging benchmark. Moreover, simply narrowing the visual field through gaze cropping proves insufficient for fully leveraging gaze information with existing vision encoders.

## 4.4 Results on EgoGazeVQA

**Baseline results and gaze-guided intent prompting.** As shown in Table 3, our experiments reveal that existing MLLMs, including advanced commercial API-based models, struggle to accurately capture human spatial and temporal intentions. However, large MLLMs such as Qwen2.5-VL and commercial API-based models demonstrate relatively strong performance on causal reasoning tasks. This can be attributed to the fact that the options provided in causal reasoning questions are typically more detailed, making it easier for models to deduce outcomes even without access to personal gaze information. These findings suggest that global visual information for MLLM is not enough to interpret finer-grained spatial and temporal intentions. To ensure a fair comparison and provide humans with access to the same information as the models, we visualized the gaze points directly on the video frames.

When we incorporate gaze signals into MLLMs, we observe performance improvements across all three prompting strategies. However, the performance gains for smaller open-source models are relatively minor. We speculate that this is due to the limited training data and model capacity, which constrain their ability to effectively interpret gaze-related prompt instructions. In contrast, larger MLLMs, such as Qwen2.5-VL-72B, exhibit significantly greater benefits from gaze signals across all prompting strategies, suggesting that model scale and capacity play a crucial role in leveraging gaze information effectively.

Furthermore, our results indicate that among the three prompting strategies, gaze saliency maps lead to the most significant performance improvement, particularly in spatial intent understanding. In contrast, for temporal intent understanding, textual prompts yield greater gains compared to the

Table 5: Performance comparison of different MLLMs across scenarios. We list baseline accuracies for each model and report changes in performance when all models adopt a **Sequential Gaze Salience Map** prompting. All model keep 9 frames input. ↑ X indicates improvements in accuracy, ↓ X denotes a decrease, – denotes no performance change.

| MLLM | LLM | Kitchen | Living Room | Medical Area | Garage | Others | Average |
|---|---|---|---|---|---|---|---|
| *Open-source Models* | | | | | | | |
| LLaVA-Next-Video | Qwen2-7B | 53.2 ↑3.2 | 52.5 ↑3.6 | 69.1 ↓1.5 | 63.2 ↑3.5 | 50.6 ↑7.7 | 54.9 ↑3.4 |
| MiniCPM-o-2.6 | Qwen2.5-7B | 40.5 ↑12 | 31.3 ↑25.2 | 29.4 ↑25 | 49.1 ↑6.2 | 32.1 ↑19.8 | 35.9 ↑17.8 |
| InternVL2.5 | InternLM2.5-7B | 59.7 ↑1.8 | 57.1 ↑1.9 | 56.9 ↑11.2 | 71.3 ↓0.9 | 50.5 ↓1.0 | 58.3 ↑1.3 |
| Qwen2.5-VL-7B | Qwen2.5-7B | 48.8 ↑1.7 | 50.7 ↑4.8 | 63.9 ↑8.3 | 56.5 ↑2.8 | 46.7 ↑3.8 | 49.7 ↑3.7 |
| Qwen2.5-VL-72B | Qwen2.5-72B | 58.5 ↑2.5 | 60.9 ↑8.1 | 70.8 ↑1.4 | 69.4 ↑2.8 | 59.9 ↑4.9 | 60.5 ↑4.9 |
| *Closed-source Models* | | | | | | | |
| GPT-4o mini | - | 57.4 ↓0.4 | 55.5 ↑3.0 | 62.5 ↑5.6 | 68.5 ↓1.8 | 50.0 ↑4.9 | 57.0 ↑1.6 |
| Gemini 2.0 Flash-Lite | - | 53.8 ↑3.0 | 43.1 ↑13.2 | 63.9 – | 63.0 – | 46.7 ↑6.0 | 51.0 ↑6.2 |

other two strategies. This suggests that MLLMs are less effective at capturing temporal correlations across frames overlaid with gaze signals, yet can more effectively reason about temporal relationships through textual gaze information. We speculate that this limitation arises from the scarcity of gaze-related visual training data, especially with temporal dimensions, making it challenging for MLLMs to develop a comprehensive understanding of temporal cues based on visual gaze signals.

Our benchmark also presents challenges to human subjects. However, humans consistently outperform the best-performing model (Qwen2.5-VL-72B) across all categories, particularly in spatial and temporal reasoning. This indicates that the task is limited by current model capabilities. The sizable gap indicates the opportunity for future work to better leverage gaze cues on intention understanding. We also vary the number of uniformly sampled frames fed to Qwen2.5-VL-72B and report the results in Appendix C.

Table 6: Distribution of error types (%) in QA pairs from human verification

| Error Type | Spatial | Temporal | Causal |
|---|---|---|---|
| Inaccurate | 17.89 | 21.03 | 8.05 |
| Irrelevant | 16.78 | 13.87 | 10.96 |
| Unanswerable | 15.88 | 9.84 | 6.26 |

**Per-scenario results.** In Table 5, Our results show that Qwen2.5-VL-72B achieves the highest overall score across various environments. Notably, both the Medical Area and Garage scenarios yield relatively high average scores. Upon examining sampled examples, we infer that these settings contain fewer objects, simplifying the task for the models. In contrast, the Kitchen scenario—characterized by higher complexity and a more cluttered environment—significantly reduces the accuracy of all tested models. This suggests that even the most advanced MLLMs still face substantial challenges in handling complex and cluttered scenes. Interestingly, gaze information enhances performance in almost all scenarios. However, some models exhibit performance regression in Medical scenarios. We found that this is due to the simplicity of objects in these environments combined with the more erratic gaze movements typical in medical scenarios.

**Per-activity evaluation.** In Table 7, we evaluate the models' capabilities across different activity types and find that performance is generally higher for medical- or health-related activities, aligning with our earlier observations on medical scenarios. In contrast, performance drops significantly in social gaming activities, where multi-person social interactions are more prevalent. These suggest that models struggle to effectively capture and interpret interpersonal dynamics from a first-person perspective, highlighting substantial opportunities for improvement in handling complex, socially oriented content.

Moreover, incorporating gaze signals results in only minor gains for medical and creative activities. For medical tasks, this is likely because intentions are often inferred from pronounced, purposeful body movements rather than gaze alone. In the case of creative activities, the frequent and drastic gaze shifts typical during multi-person interactions introduce additional challenges for MLLMs interpret gaze-related prompts accurately. These observations underscore the need for more refined strategies to leverage gaze information effectively in diverse and dynamic scenarios.

Table 7: Performance comparison of different MLLMs across activities. We list baseline accuracies for each model and report changes in performance when all models adopt a **Sequential Gaze Salience Map** prompting. All model keep 9 frames input. ↑ X indicates improvements in accuracy, ↓ X denotes a decrease, – denotes no performance change.

| MLLM | LLM | Cooking | Gaming | Health | Mechanics | Creative | Others |
|------|-----|---------|--------|--------|-----------|----------|--------|
| *Open-source Models* | | | | | | | |
| LLaVA-Next-Video | Qwen2-7B | 54.3 ↑3.1 | 51.4 ↑4.2 | 71.7 ↓1.9 | 63.2 ↑3.5 | 54.4 ↑3.5 | 50 ↑6.0 |
| MiniCPM-o-2.6 | Qwen2.5-7B | 40.9 ↑11.8 | 31.1 ↑25 | 32.1 ↑24.5 | 49.1 ↑6.2 | 28.1 ↑2.6 | 32.7 ↑19.1 |
| InternVL2.5 | InternLM2.5-7B | 60.0 ↑2.1 | 58.6 ↑3.2 | 62.7 ↑6.8 | 70.4 – | 49.3 ↑4.4 | 48.9 – |
| Qwen2.5-VL-7B | Qwen2.5-7B | 49.6 ↑1.3 | 55.7 ↑4.2 | 66.1 ↑3.4 | 55.6 ↑3.7 | 50.7 ↑1.5 | 36.7 ↑8.5 |
| Qwen2.5-VL-72B | Qwen2.5-72B | 58.7 ↑2.1 | 65.3 ↑10.5 | 71.2 ↑3.4 | 69.4 ↑7.5 | 52.3 ↑2.8 | 52.1 ↑6.4 |
| *Closed-source Models* | | | | | | | |
| GPT-4o mini | - | 57.4 ↓0.6 | 57.3 ↑4.5 | 64.4 ↑6.8 | 67.6 ↓1.9 | 52.2 ↑4.3 | 48.9 ↑2.2 |
| Gemini 2.0 Flash-Lite | - | 53.8 ↑3.4 | 45.5 ↑14.7 | 66.1 – | 63.0 – | 52.2 ↑2.9 | 40.4 ↑5.9 |

## 4.5 Empirical study of LoRA fintuning

**Cross-dataset LoRA fine-tuning.** We LoRA-fine-tune Qwen2.5-VL-7B with LLaMA-Factory [38], and report the results in Table 8. We first train the adapter on the EGTEA split (about 500 gaze-guided QA pairs) and evaluate on the remaining Ego4D + EgoExo data (1200 QA pairs). Next, we reverse the direction, fine-tuning on Ego4D + EgoExo and testing on EGTEA, to examine cross-domain transfer. For reference, we also include zero-shot results from Qwen2.5-VL on the same test sets. All evaluations are conducted without gaze prompts.

Table 8: LoRA fine-tuning across datasets with different scale training splits.

| MLLMs | Methods | Test Data | Spatial | Temporal | Causal | Average |
|-------|---------|-----------|---------|----------|--------|---------|
| Qwen2.5-VL-7B | Zero-shot | Ego4D + EgoExo | 40.4 | 43.3 | 78.2 | 54.0 |
| | SFT on EGTEA | Ego4D + EgoExo | 67.7 | 56.5 | 84.4 | 69.5 |
| | Zero-shot | EGTEA | 45.7 | 32.9 | 77.3 | 52.0 |
| | SFT on Ego4D + EgoExo | EGTEA | 66.5 | 47.0 | 82.8 | 65.4 |

Notably, the LoRA model outperforms the zero-shot baseline whether the adapter is trained on the small EGTEA split or on the larger Ego4D + EgoExo split. All categories improve, with the most substantial gain in spatial reasoning at 67.7%, indicating that a modest amount of gaze-conditioned QA quickly aligns the model's attention to object-centric cues. These results suggest that limited domain-specific data can already benefit light MLLMs to better leverage the exlicity gaze signas provided in the prompt.

## 4.6 Additional analsyis

Table 9: Prompt-based gaze estiamtion with MLLM (Qwen2.5 VL-72B) and effect on EgoGazeVQA. GT: Ground Truth. GS: Gaze Estimation.

| Datasets | w/o | GT | $\Delta$(GT − w/o) | GE | $\Delta$(GE − w/o) | MSE |
|----------|-----|-----|-------------------|-----|-------------------|-----|
| Ego4D | 60.1 | 67.0 | +6.9 | 60.5 | +0.4 | 0.154 |
| EgoExo | 61.7 | 67.3 | +5.6 | 64.5 | +2.8 | 0.038 |
| EGTEA | 59.3 | 64.0 | +4.7 | 58.7 | -0.6 | 0.119 |
| Average | 60.5 | 66.3 | +5.8 | 61.5 | +1.0 | 0.099 |

**Prompted MLLM gaze estimation and impact on EgoGazeVQA.** To further show that explicit gaze signal is vital for MLLM to understand human intent, we further query Qwen2.5-VL-72B for egocentric gaze estimation using the following prompt "*Mark the coordinates the camera wearer is looking at*""; the returned $(x, y)$ forms a frame-wise gaze map that is fed back as a sequential gaze salience map prompt (GazeS) for EgoGazeVQA.

Table 9 shows that gaze estimation accuracy is strongly correlated with performance improvement. (for example, MSE = 0.038 in EgoExo), the evaluation results lags slightly behind those obtained using the ground-truth gaze. In contrast, when gaze estimation is less accurate, the benefit from GE diminishes drastically and may even slightly hurt performance.

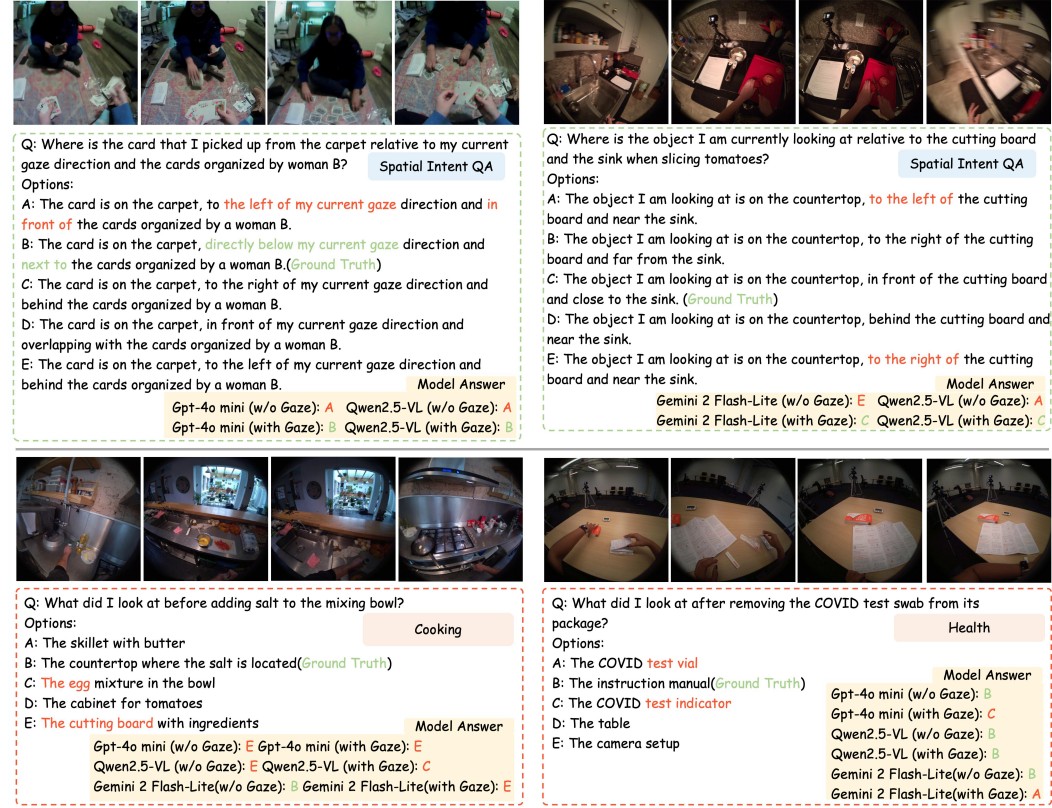

Figure 4: Visual examples from our EgoGazeVQA benchmark and gaze-guided prompting

## 4.7 Analysis of visual results

We provide visualization of gaze-guided prompting results on our benchmark in Figure 4. As shown in the successful examples in the first rows, gaze signals significantly enhance MLLMs' performance in tasks demanding precise spatial reasoning and intent interpretation, such as distinguishing closely situated objects in the kitchen and accurately inferring user focus in complex scenes. However, gaze signals fail to improve performance when there is drastic body motion, as illustrated by the left example in the last row. This suggests that MLLMs struggle to understand the egocentric video with heavy camera motion even with the help of gaze signals. Additionally, gaze saccades can mislead the model into focusing on irrelevant objects, further limiting the effectiveness of gaze cues.

## 5 Conclusion

In this paper, we introduced EgoGazeVQA, an egocentric gaze-guided video question answering benchmark designed to evaluate the ability of existing MLLMs to interpret user intentions from first-person videos. Additionally, we proposed and examined gaze-guided prompting methods that utilize gaze information as a direct indicator of user focus and intent, enabling a deeper understanding of spatial, temporal, and causal relationships in egocentric contexts. Our experiments show that integrating gaze cues significantly enhances MLLM performance across various scenarios and activities. We believe that EgoGazeVQA represents an important step towards developing proactive and personalized AI assistants with egocentric MLLMs. Furthermore, our work opens up promising directions for research on multi-sensory MLLMs and MLLMs explainability. Moreover, by grounding MLLMs in human-perspective signals like gaze, our work also provide foundations for cobodied intelligence.

# 6 Acknoledgement

This work is supported by Beijing Natural Science Foundation (L242019). We thank Bosong Qi, Yixuan Sun, Yan Li, Jinru Liu, Sijia Liu, Xiaolong Wu, and Zhenghui Sun for their help with manual verification to improve the quality of the QA pairs. We also thank Yiwei Bao and Zhimin Wang for helpful discussions on ideas. Finally, we thank our reviewers for their valuable comments.

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

# A    Appendix Overview

Our supplementary includes the following sections:

- **Section B: Framework details.** Details for pipline construction and prompt strategy implementation.
- **Section C: More experiment results.** Additional performance evaluation and performance analysis.
- **Section D: Limitations.** Discussion of limitations and future work in our paper.
- **Section E: Prompt design.** Prompt for generating the EgoGazeVQA dataset and evaluating the performance.

We have shared the following artifacts:

| Artifcat | Link | License |
|---|---|---|
| Code Repository | https://github.com/taiyi98/EgoGazeVQA | Apache-2.0 license |
| Data | https://huggingface.co/datasets/taiyi09/EgoGazeVQA | CC BY 4.0 |

# B    Framework details

**Cross-dataset LoRA fine-tuning details.**   We fine-tune Qwen2.5-VL-7B-Instruct with a rank-16 LoRA adapter (*target = all layers*) in llama-factory. Two data regimes are considered: (i) the small EGTEA split (500 QA pairs) and (ii) the larger Ego4D + EgoExo split (1 200 QA pairs). Both runs use a batch size of 8, three epochs, a cosine schedule with peak learning rate $1.5 \times 10^{-4}$ (10 % warm-up), and bfloat16 precision; training is carried out on a single NVIDIA A100-80 GB.

EGTEA. Training finishes in 2.5 hours, reaches a final loss of 0.556, and consumes $2.1 \times 10^{14}$ FLOPs. Ego4D + EgoExo. Training takes 6 hours, attains a lower loss of 0.341, and accounts for $4.7 \times 10^{14}$ FLOPs.

**Sequential gaze salience map init.** To keep one unified prompt for both APIs-based and on-device MLLMs, we compress the entire gaze trace into a single salience heat-map. Fixations are weighted in reverse time order—later frames receive higher intensities—so the map mirrors the human bias to rely more on recent visual information when answering a question.

---

**Algorithm 1** Sequential Gaze Salience Map Generation

---

**Require:** Image sequence $I$, Gaze data $G$
**Ensure:** Salience map $S$
 1: Initialize $S \leftarrow 0$                                                ▷ Zero matrix with image dimensions
 2: **for** $g_i \in G$ **do**
 3:      $(x, y) \leftarrow \text{Scale}(g_i, I)$
 4:      **for** $(dx, dy) \in \text{Grid}(-r, r, \text{step} = 10)$ **do**
 5:          **if** $\sqrt{dx^2 + dy^2} \leq r$ **then**
 6:              $S(y + dy, x + dx) \mathrel{+}= \omega \cdot \exp(-\frac{dx^2 + dy^2}{2\sigma^2})$
 7:          **end if**
 8:      **end for**
 9:      $S(y, x) \mathrel{+}= \omega \cdot \text{Weight}(i)$
10: **end for**
11: $S \leftarrow \text{GaussianBlur}(S, k = 6\sigma + 1)$
12: Normalize $S \rightarrow [0, 255]$
13: **return** $S$

---

## C More experiments results

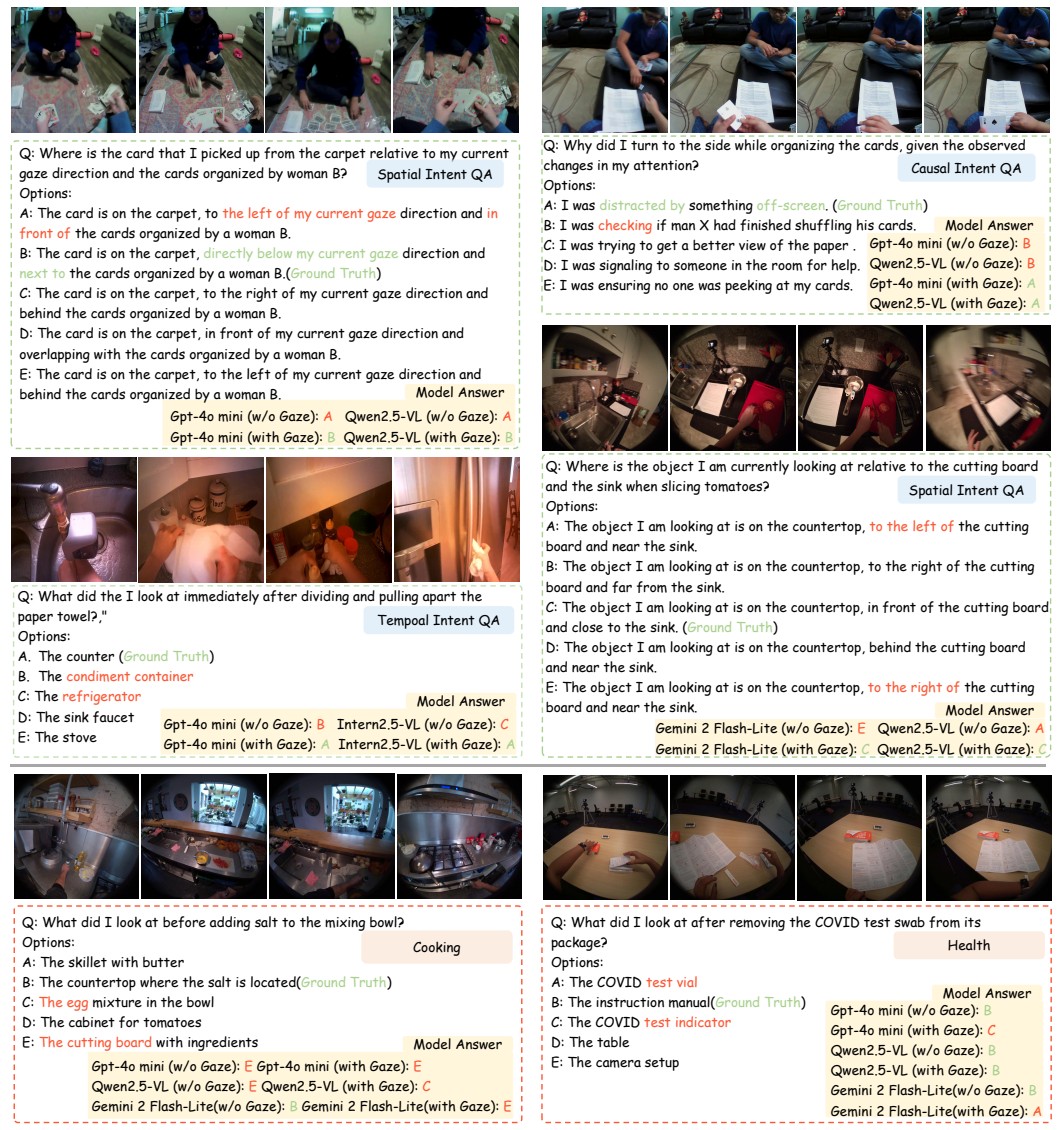

Figure 5: We present examples from our EgoGazeVQA benchmark, illustrating how gaze information influences model predictions. Each example includes the question, multiple-choice options, and the ground truth, with correct and incorrect predictions highlighted for from prevailing MLLMs with and without gaze-guided prompting.

### C.1 Effect of varying numbers of input frames.

Since MLLMs perceive video through sparsely sampled images, we vary the number of uniformly sampled frames fed to Qwen2.5-VL-72B and report the results in Table 10. Notably, adopting 9 frames as the default setting offers the best trade-off between accuracy and computational cost.

### C.2 Why we report MSE for prompt-based gaze estiamtion.

Most egocentric gaze-estimation[18][14] output a *heat-map* and therefore adopt hit/miss metrics such as F1 or AUC within a fixed angular tolerance. In contrast, our prompt returns an explicit $(x, y)$ coordinate, not a density map. For a single point prediction the most

Table 10: Effect of varying numbers of input frames

| Frames | Spatial | Temporal | Causal | Average |
|--------|---------|----------|--------|---------|
| 4 | 46.9 | 45.5 | 78.6 | 57.0 |
| 6 | 47.4 | 45.0 | 79.0 | 57.1 |
| 9 | 47.9 | 44.0 | 79.8 | 57.2 |
| 14 | 49.2 | 43.3 | 79.0 | 57.1 |
| 18 | 47.2 | 42.6 | 79.2 | 56.3 |

natural error measure is the Euclidean distance to the ground-truth fixation, which—after normalising image size—reduces to a mean-squared error (MSE). Using MSE avoids choosing an arbitrary threshold, is scale-independent, and directly reflects how far the predicted fixation deviates from the true one; hence we employ it in Table 9.

## D    Limitation and Future Work

**Limitations.** Our benchmark has several limitations that warrant consideration. First, most egocentric videos with eye-tracking data are recorded in indoor scenarios, which introduces a potential benchmark bias and limits the generalizability of the results to outdoor or more diverse environments. Second, we did not explore the audio modality in this benchmark, despite its significant potential in social scenarios. This decision was influenced by the limited availability of socially oriented test sets and the absence of audio signals in certain video sources, such as EGTEA Gaze+. Third, we did not evaluate model performance across a diverse set of video lengths, as this was not the primary focus of this paper. The current evaluation primarily relies on short to medium-length videos, which may overlook potential challenges and model limitations associated with processing longer videos, such as maintaining temporal coherence and handling information overload.

**Future Work.** This benchmark opens up several promising research directions. For instance, exploring how gaze signals can prune the input visual tokens of MLLMs—analogous to how the human primary visual cortex processes visual information. Furthermore, investigating methods to prompt MLLMs to infer gaze signals implicitly, without relying on explicit eye-tracking data, could significantly enhance their generalizability in egocentric settings. Moreover, this benchmark can be valuable for MLLMs explanabilty study, by aligning model attention with human gaze patterns.

## E    Prompt design

### E.1    Full Prompt for Benchmark Construction

#### E.1.1    Spatial QA Generation

You are an expert in understanding gaze information and spatial localization in ego-centric video data. Your task is to generate spatial localization and gaze-aware video QA benchmarks. These benchmarks are designed to evaluate nuanced understanding of gaze-based spatial relations in video scenes.

**Provided Inputs:**

**RGB keyframe**: A visual snapshot of the scene captured from the first-person perspective.

**Keyframe caption**: A textual description of the keyframe content, including objects and their spatial arrangement.

**Ego-centric gaze information**: Includes gaze fixation points.

**Requirements:**

- **Create Question**: Formulate spatial localization questions based on the given input with a strong emphasis on the ego-centric gaze. Questions should incorporate both gaze dynamics and spatial relationships in the scene, such as:
  The object's position relative to the gaze direction.
  Combining gaze focus and object-to-object spatial relations.

- **Generate Answer Options**: Provide five plausible answer options for each question. Ensure that:

Only one option is correct.

The other options are plausible but incorrect, requiring nuanced understanding of gaze fixation, object relationships, and spatial layout to differentiate.

- **Focus on Detailed Spatial and Gaze-Based Relations**: Unlike traditional benchmarks with simple spatial answers (e.g., "on the table"), your answers should include detailed gaze-based spatial relationships (e.g., "On the table, to the right of the object I looked at for the longest time"). This tests the model's ability to interpret both gaze data and spatial relations.

**Example:**

**Question:** What is the relative relationship between the knife and my current fixation?

**Answer Options:**

- A: The knife is on the countertop, to the left of my current fixation.
- B: The knife is near the edge of the countertop, behind my current fixation.
- C: The knife is near the edge of the countertop, to the right of my current fixation.
- D: The knife is on the countertop, in front of my current fixation.
- E: The knife is near the edge of the countertop, to the left of the cutting board and my current fixation.

**Correct Answer:** C.

### E.1.2 Temporal QA Generation

You are an expert in contextual temporal reasoning for videos. Your task is to generate high-quality Contextual Temporal Reasoning VideoQA benchmarks that emphasize event-based responses. Unlike traditional benchmarks that rely on time-based answers (e.g., "from 10s to 50s in the video"), this task focuses on generating questions and answers based on event relationships within the video.

**Provided Inputs:**

**RGB Keyframe**: A visual snapshot of the scene captured from the first-person perspective.

**Keyframe Caption**: A textual description of the keyframe content, including objects and their spatial arrangement.

**Ego-centric Gaze Information**: Includes gaze fixation points.

**Requirements:**

- **Create Question**
  Focus on what the viewer looked at before or after an action.Explore action sequences based on gaze transitions.Highlight cause-and-effect relationships between gaze and actions.Include gaze shifts.

- **Generate Answer Options**: Provide five plausible answer options for each question. Ensure that:
  Use natural reasoning: Options should reflect events based on gaze, not timestamps.
  Ensure answers are grounded in visible objects and actions from the frames and gaze patterns.

**Example:**

**Question:** What did I look at after I put down the knife?

**Answer Options:**

- A: The cutting board.
- B: The stove.
- C: The refrigerator.
- D: The plate on the counter.
- E: The sink.

**Correct Answer:** A.

### E.1.3 Causal QA Generation

You are an expert in gaze-based event causal understanding within ego-centric video environments. Your task is to design comprehensive Gaze-Informed Causal Reasoning ego-centric VideoQA benchmarks. The benchmarks should focus on complex, high-level reasoning that integrates gaze dynamics with event interactions.

**Input Data:** Visual: 9 first-person RGB frames + captions. Ego-centric gaze information: Includes gaze fixation points.

**Event Parsing:** Identify action chains (e.g., pick glass → drink → glance at bottle → pour water). Link actions to gaze patterns, focusing on implicit cause-and-effect relationships.

**Gaze Dynamics:** Cluster gaze points as natural scene regions (e.g., 'cup rim', 'bottle cap area'). Track gaze trajectory shifts (e.g., 'suddenly locked onto...'). Focus on: Gaze as a predictive signal for upcoming actions; Ambiguous gaze paths that may point to multiple plausible behaviors.

**Requirements:**

- **Ego-centric Question Construction**: Create 5 multiple-choice options: 1 correct and 4 misleading. Use templates like: Why did [Subject] perform [Action] while [Ongoing Task], given the observed changes in my attention? Why did [Subject] [Action] while [Other Subject] was [Action], considering the shifts in my attention? What was [Subject] trying to achieve by [Action], based on the changes in my attention during [Task]?

- **Generate Answer Options** Ensure that: correct answers require concurrent gaze features to explain actions. Avoid using external contextual clues that are not related to gaze. Express temporal relationships through event sequences, but avoid explicit time references (e.g., 'after Frame 3', 'during the first phase'). Focus on my gaze's role in anticipating or influencing actions, but avoid overly simplistic or surface-level reasoning.

- **Distractor Design**: Include reverse-causal options, where my gaze behavior is misinterpreted as being a result of the action; Spatial-proximity traps where objects in close proximity are incorrectly linked to my gaze behavior; High-salience distractors that divert attention to irrelevant but visually prominent elements; Create social influence traps

**Example Question:**

Why did I shuffle the cards while organizing them, given the observed changes in attention?

**Answer Options:**

- A: I was focusing on the deck to ensure it was properly shuffled.
- B: I was distracted by the cards' edges and kept adjusting them.
- C: I was checking if any specific card was missing from the deck.
- D: I was trying to hide certain cards from the others.
- E: I was preparing the cards for a trick.

**Correct Answer:** C.

### E.2 Full Prompt for Gaze-Guide Prompting

### E.2.1 Gaze as Textual Prompt

**INPUT:**

Gaze information for the relevant frames:

- Frame [*i+1*]: Gaze(gaze_x, gaze_y)

I provide you with a video and the normalized gaze coordinates for each corresponding frame. You need to follow these steps to answer the questions:

(a) Observe the position of the annotated gaze points in each frame. The coordinate is from left to right for the x-axis and from top to bottom for the y-axis.

(b) Analyze the video while considering the gaze point information and then answer the questions.

(c) **Question:** [*question*] **Options:** [*answer_options*]

Choose the most appropriate option. Return the letter of the correct option.

### E.2.2 Gaze as Visual Prompt

**INPUT:**

I provide you with a video that contains gaze information. For each frame, the gaze point will be marked on the image with a red circle spot. Choose the correct option based on the first-person perspective scene question.

You must follow these steps to answer the question:

(a) Focus on the objects marked by each red circle spot.

(b) Observe the video chronological order according to the gaze sequence in step 1.

(c) **Question:** [*question*] **Options:** [*answer_options*]

Choose the most appropriate option. Return the letter of the correct option.

### E.2.3 Gaze Salience Maps as Prompt

**INPUT:**

I provide you with a Picture Frame [*0*] and a video Frame [*1-9*]. Choose the correct option based on the first-person perspective scene question.

You must follow these steps to answer the question:

(a) Frame [*0*] is the saliency heatmap of the gaze trajectory from the first-person perspective, with gaze sequence from dark spot to bright spot.

(b) Remember the location and time sequence of gaze areas in Frame [*0*].

(c) Observe the video according to the gaze sequence in step 2.

(d) **Question:** [*question*] **Options:** [*answer_options*]

Choose the most appropriate option. Return the letter of the correct option.

