# OpenReview forum: "In the Eye of MLLM: Benchmarking Egocentric Video Intent Understanding with Gaze-Guided Prompting"
_NeurIPS.cc/2025/Datasets_and_Benchmarks_Track — NeurIPS 2025 Datasets and Benchmarks Track poster_

### Official Review · Reviewer_j14H · 2025-06-26

**Rating:** 4
**Confidence:** 4

**Summary:**

To address the issue of previous benchmarks neglecting the role of gaze in egocentric video understanding, this paper presents a new benchmark named EgoGazeVQA, which consists of egocentric gaze-guided video question answering. In addition, the authors also propose a gaze-guided intent prompting method to leverage spatial, temporal, and intent-related cues.

**Additional Feedback:**

For suggestions, I think the authors should improve the dataset size of EgoGazeVQA to better capture the diversity and complexity of real-world egocentric gaze behaviors.

I have no additional questions.

**Dataset Code Accessibility:**

Yes

**Dataset Code Comments:**

The dataset code is available.

**Ethical Considerations:**

No, there are no or only very minor ethics concerns

**Final Justification:**

The authors' response has addressed the majority of my concerns.
Therefore, I raise my score to "Borderline accept".

**Limitations Weaknesses:**

1. The proposed EgoGazeVQA benchmark is limited in data size. For example, EgoGazeVQA only has 1757 questions (1.7k), which is less than EgoSchema (5k) and EgoTextVQA (7k). This relatively small scale may hinder the robustness of the evaluation and limit the generalizability of models. The authors should include more diverse and comprehensive question-answer pairs.

2. This paper only provides some prompting strategies for this task, and there is no simple baseline to address the challenges in EgoGazeVQA.

3. In the QA pair generation step, is there a risk of hallucination or factual inaccuracies propagating into the QA pairs? Have the authors evaluated the proportion of such errors?

**Strengths Contributions:**

1. This paper first exploits egocentric gaze-guided video intent QA in the egocentric video understanding community.

2. This paper investigates gaze-guided prompting paradigms that leverage cues from spatial, temporal, and intent levels.

3. The authors present empirical analysis to validate the effectiveness of gaze signals in understanding human intentions in the egocentric field.

---

> ### Author Rebuttal · Authors · 2025-07-31
>
> We sincerely thank Reviewer j14H for constructive feedback, which have helped us improve our work. In response, we offer additional experimental results and explanations to address the suggestions raised by Reviewer j14H.
>
> **Q1: The proposed EgoGazeVQA benchmark is limited in data size.**
>
> We appreciate the reviewer’s comment on the benchmark size. We would like to point out that EgoTextVQA and EgoSchema do not require gaze tracking data. Since only a small portion of existing egocentric videos datasets contain high-quality gaze tracking data, it is challenging to scale up the benchmark size without introducing duplication. Despite such constraints, our EgoGazeVQA considers a richer collection of videos (913) compared to EgoThink (700) and EgoMemoria (629). Moreover, our dataset size (1757 QA pairs) is comparable to widely recognized video understanding benchmarks like LVBench[1] (1549 QA pairs) and MMBench-Video[2] (1998 QA pairs).
>
> **Q2: There is no simple baseline to address the challenges in EgoGazeVQA.**
>
> We thank the reviewer for the valuable suggestion to include additional baselines. In Table 3 of the main paper, we report performance of representative MLLMs on EgoGazeVQA to establish demonstrative baselines and highlight the significant room for improvement on gaze-relevant tasks. Here, we provide results using CLIP (ViT-H-14) and EgoVLP as simple baselines, where we compute the cosine similarity between the visual embeddings of the input and the text embeddings of each answer option, selecting the one with the highest similarity as the prediction. We also evaluate BLIP-2, where generates answers based on the visual embeddings and input question. We calculate the cross-entropy loss between the generated answer and each of the candidate options, and select the one with the lowest loss as the most likely match.
>
> For both baselines, we follow the same video preprocessing pipeline described in the main paper. To mimic human visual attention, we use gaze fixation as an attentional map to select around 20\% of each frame, resulting in what we term the GazeMap variant (i.e., CLIP-GazeMap, EgoVLP-GazeMap and BLIP-2-GazeMap), with all other components unchanged.
>
> | Method         |Similarity |  Performance |
> |----------------|:------:|:-------:|
> | Human  |N/A |  76.8  |
> | CLIP ViT-H-14  |Video-A |  21.63  |
> |   CLIP-GazeMap |Video-A |  21.79  |
> |     EgoVLP     |Video-A |  22.95  |
> | EgoVLP-GazeMap |Video-A |  22.65  |
> |     BLIP-2     |  N/A   |  27.60  |
> | BLIP-2-GazeMap |  N/A   |  27.21  |
> |Qwen2.5-VL-72B  |  N/A   |  60.5   |
> |Qwen2.5-VL-72B-GazeVisual | N/A| 63.9  |
>
>
> Notably, these simple baselines perform only marginally better than random guessing and fall far short of human expert performance, indicating that CLIP-based methods struggle to tackle this challenging benchmark. Moreover, simply narrowing the visual field through gaze cropping proves insufficient for fully leveraging gaze information with existing vision encoders. This reinforces the need for the specialized prompting strategies proposed in our work, which incorporate spatiotemporal and intent-aware cues. We will incorporate this dicsusion in our main paper.
>
> **Q3: Is there a risk of hallucination or factual inaccuracies propagating into the QA pairs?**
>
> We agree with the reviewer that generating QA pairs using MLLMs introduces a potential risk of hallucinations. To mitigate this, we first applied automatic filtering to remove duplicate or malformed QA pairs, followed by rigorous human verification. All QA pairs were manually reviewed along six dimensions: Relevance, Answerability, Fluency, Accuracy, Conciseness, and Difficulty.
>
> The three most common types of errors identified during human review are summarized below (Note that each generated QA pair may have more than one type of human verification error):
>
> | Category | Inaccurate(\%) | Irrelevant (\%) | Unanswerable(\%) |
> |------|:-----:|:---------:|:------:|
> |  Saptial   |  17.89  |  16.78     |   15.88   |
> | Temporal   |  21.03 |   13.87   |  9.84   |
> |   Causal  |    8.05 |     10.96   |    6.26   |
>
> For QA items identified with such issues, we manually corrected them. If a sample was too ambiguous to fix , we removed it entirely. This ensures the overall quality and reliability of the final dataset.  We will add the statistics to our main paper.
>
> We hope our response sufficiently addresses your comments. If you have any further questions or concerns, please feel free to reach out to us for further discussion.
>
> **Reference**
>
> [1] Wang, Weihan, et al. "Lvbench: An extreme long video understanding benchmark." arXiv preprint arXiv:2406.08035 (2024).
>
> [2] Fang, Xinyu, et al. "Mmbench-video: A long-form multi-shot benchmark for holistic video understanding." Advances in Neural Information Processing Systems 37 (2024): 89098-89124.

---

> > ### Comment · Reviewer_j14H · 2025-08-04
> > **Response to rebuttal**
> >
> > Thanks for the authors' response. It has addressed the majority of my concerns. However, I still have concerns about the size of the proposed dataset.

---

> > > ### Author Response · Authors · 2025-08-05
> > > **Clarification on Dataset Size and Benchmark Comparisons**
> > >
> > > Dear Area Chairs and Reviewer j14H,
> > >
> > > Thank you again for your follow-up comments and suggestion. We understand your concern regarding dataset size, and would like to provide further context by comparing EgoGazeVQA to recent peer-reviewed video understanding benchmarks:
> > >
> > > | Benchmark | Type |  Videos   | QA pairs |Citations | Year | Venue |
> > > |-----------|:--------:| :--------:|:---------------:|:--------:|:--------:|:--------:|
> > > | Video-MME [1] | Exo |   900    |   2,700  |  478     |  2025           | CVPR
> > > | MMWorld [2]   | Exo |   1,910    |   1,599    |   28      |   2025     | ICLR|
> > > | Vidcomposition [3] | Exo |   982   |    1,706  |    9  |     2025       | CVPR|
> > > | Q-Bench-Video [4]| Exo |  1,800| 2,378 |  9    |  2025 | CVPR|
> > > | Lvbench [5] | Exo |   500   |    1,549   |    92  |     2024       | ICCV|
> > > | Mmbench-video [6] | Exo   | 600  | 1,998  | 93 | 2024 | NeurIPS |
> > > | VideoMindPalace [7]   | Ego   | 200  | 1,800  | 1 | 2025 | CVPR |
> > > | EgoThink [8]      | Ego |  700  |  700 | 34 | 2024  | CVPR |
> > > | **EgoGazeVQA(Ours)** | Ego | 913 | 1,757 | / | 2025| / |
> > >
> > > As the table shows, our dataset EgoGazeVQA is within the same scale as recent peer-reviewed benchmarks. Moreover, Unlike most existing benchmarks, each QA pair in EgoGazeVQA is associated with  **gaze eye-tracking data**, which provides a valuable cue for MLLMs to better infer the user’s intent in egocentric settings.
> > >
> > > We hope these clarifications address your concern. Thank you again for your valuable comments and thoughtful engagement!
> > >
> > > Best,
> > >
> > > The Authors
> > >
> > > **Reference**
> > >
> > > [1] Fu, Chaoyou, et al. "Video-mme: The first-ever comprehensive evaluation benchmark of multi-modal llms in video analysis." Proceedings of the Computer Vision and Pattern Recognition Conference. 2025.
> > >
> > > [2] He, Xuehai, et al. "MMWorld: Towards Multi-discipline Multi-faceted World Model Evaluation in Videos." The Thirteenth International Conference on Learning Representations.
> > >
> > > [3] Tang, Yunlong, et al. "Vidcomposition: Can mllms analyze compositions in compiled videos?." Proceedings of the Computer Vision and Pattern Recognition Conference. 2025.
> > >
> > > [4] Zhang, Zicheng, et al. "Q-Bench-Video: Benchmark the Video Quality Understanding of LMMs." Proceedings of the Computer Vision and Pattern Recognition Conference. 2025.
> > >
> > > [5] Wang, Weihan, et al. "Lvbench: An extreme long video understanding benchmark." arXiv preprint arXiv:2406.08035 (2024).
> > >
> > > [6] Fang, Xinyu, et al. "Mmbench-video: A long-form multi-shot benchmark for holistic video understanding." Advances in Neural Information Processing Systems 37 (2024): 89098-89124.
> > >
> > > [7] Huang, Zeyi, et al. "Building a Mind Palace: Structuring Environment-Grounded Semantic Graphs for Effective Long Video Analysis with LLMs." Proceedings of the Computer Vision and Pattern Recognition Conference. 2025.
> > >
> > > [8] Cheng, Sijie, et al. "Egothink: Evaluating first-person perspective thinking capability of vision-language models." Proceedings of the IEEE/CVF Conference on Computer Vision and Pattern Recognition. 2024.

---

> > > > ### Comment · Reviewer_j14H · 2025-08-05
> > > > **Response**
> > > >
> > > > Thank you for your feedback. I am inclined to raise my score to "Borderline accept".
> > > >
> > > > I strongly recommend expanding the size of the proposed dataset in future research.

---

> > ### Author Response · Authors · 2025-08-05
> >
> > Dear Reviewer j14H,
> >
> > Thank you for taking the time to review our responses and for your recognition of our work. We appreciate your suggestion about expanding the dataset size.  Larger and more diverse benchmarks are important for the field. They help prevent overfitting and allow better evaluation of models. We will keep this suggestion in mind for future work.
> >
> > We will add the dataset comparison table in the supplementary material for clearer discussion and analysis.
> >
> > Thanks again for your insightful comments.
> >
> > Best regards,
> >
> > The Authors

---

### Official Review · Reviewer_p3sj · 2025-07-01

**Rating:** 4
**Confidence:** 3

**Summary:**

This paper introduces "EgoGazeVQA," a novel benchmark for evaluating Multimodal Large Language Models (MLLMs) in understanding user intent from egocentric videos by incorporating gaze signals. Egocentric videos capture first-person perspectives, aligning visual perception, attention, and context, but existing benchmarks overlook gaze as a key indicator of intent. It addresses this gap by leveraging gaze data to generate spatial, temporal, and causal intent question-answering (QA) tasks.

**Dataset Code Accessibility:**

Yes

**Dataset Code Comments:**

Code and data can be seen in Appendix A.

**Ethical Considerations:**

No, there are no or only very minor ethics concerns

**Final Justification:**

Thanks for the authors' response. I will raise to "borderline accept" rating.

**Limitations Weaknesses:**

1. Data and Generalizability Limitations: The benchmark relies heavily on indoor scenarios, introducing bias and limiting generalizability to outdoor or dynamic environments. The focus on short to medium-length videos overlooks challenges in long-video processing, such as maintaining temporal coherence, which may reduce the benchmark's applicability to real-world streaming data.

​2. Implementation Gaps: The gaze-guided prompting strategies show performance gains but have inconsistencies; for example, salience maps excel in spatial intent but underperform in temporal tasks due to MLLMs' poor temporal correlation handling. The paper does not thoroughly address how gaze estimation inaccuracies impact prompting effectiveness, leaving a gap in error analysis.

**Strengths Contributions:**

​1. Novelty and Significance: EgoGazeVQA is the first benchmark to integrate gaze signals for egocentric video intent understanding, filling a critical gap in MLLM evaluation. By focusing on spatial, temporal, and causal intent QA, it enables more personalized AI assistants, which is timely given the rise of wearable devices.

2. Rigorous Methodology: The benchmark construction is robust, using diverse datasets and human verification for QA pair quality. The experiments are comprehensive, covering multiple MLLMs and scenarios, with detailed metrics like accuracy and MSE.

3. Practical Impact and Openness: The work has high practical value for developing intent-aware AI systems, with real-world applications in healthcare, gaming, and daily assistance. The authors provide full access to code and data, including detailed prompts and evaluation scripts, facilitating reproducibility and community adoption.

---

> ### Author Rebuttal · Authors · 2025-07-31
>
> We thank the reviewer p3sj for recognizing the novelty, practical value and concerns. Below we address the two remaining concerns.
>
> **Q1: Indoor bias and short‑video focus.**
>
> We thank the reviewer for raising concerns regarding the generalizability of our benchmark. We provide further clarification:
>
> * Limited outdoor gaze data due to hardware constraints. Current head‑mounted eye‑trackers (e.g., HoloAssist, Aria, Tobii Pro) depend on near‑infrared (NIR) illumination and the pupil–corneal reflection. In outdoor scenes, sun light and glasses can seriously disturb the reflective properties of IR light. Methods using IR can therefore be less reliable in these situations[1], while pupil constriction further reduces the signal. As a result, gaze tracking in outdoor scenes suffers from significant inaccuracies. These limitations are the reason existing egocentric datasets (e.g., Ego4D, EgoExo, EGTEA, HoloAssist-Aria) do not include outdoor gaze annotations.
>
> * Short and medium-length videos (30s–2min) has been widely used in many egocentric activity understanding dataset (AssistQ[2], EgoSchema[3], EgoTextVQA[4]) and video language model benchmark (ActivityNet-QA[5], MMBench-Video[6]). Although long video understanding with gaze context is a promising direction, limitations the scarcity of egocentric video data with accurate eye-tracking signals, and high computational cost present additional challenges for academic research. We plan to further explore this direction in future work.
>
> We will properly discuss these constraints upon the release of our benchmark.
>
> **Q2: Inconsistencies on performance improvement and impact of gaze‑estimation error.**
>
> Thank you for your suggestion. We acknowledge that the performance gains from gaze-guided prompting exhibit some inconsistencies. This actually highlights the challenging nature of our benchmark, which cannot be easily solved with gaze-guided prompting alone, and underscores the need for further attention and investigation from the community.
>
> We also conducted a more detailed analysis at both the dataset level and category level. GT refers to using gaze ground truth in our prompting strategy, while GE refers to using gaze estimation.
>
> | Datasets |  w/o(baseline) | GT | $\Delta$ (GT-w/o) |  GE | $\Delta$(GE-w/o) | MSE   |
> |---------|:----:|:--:|:-----------------:|:---:|:----------------:|:-----:|
> | Ego4D   | 60.1 | 67.0  | +6.9          | 60.5  | +0.4          | 0.154 |
> | EgoExo  | 61.7 | 67.3  | +5.6          | 64.5  | +2.8          | 0.038 |
> | EGTEA   | 59.3 | 64.0  | +4.7          | 58.7  | -0.6          | 0.119 |
> | Avg.    | 60.5 | 66.3  | +5.8          | 61.5  | +1.0          | 0.099 |
>
> In datasets level table, we see that in EGTEA, where gaze estimation is less accurate (MSE = 0.119), gaze-guided prompting does not improve performance and can even slightly hurt it. In contrast, when gaze prediction is accurate in EgoExo (MSE = 0.038), it brings a clear performance boost.
>
> | Category | w/o(baseline)  | GT   | $\Delta$(GT - w/o) | GE   | $\Delta$(GE - w/o) |
> | -------- | :----: | :----: | :------------: | :----: | :------------: |
> | Spatial  | 57.1 | 64.3 | +7.2         | 53.8 | -3.3         |
> | Temporal | 45.2 | 50.3 | +5.1         | 48.6 | +3.4         |
> | Causal   | 79.4 | 84.3 | +4.9         | 82.2 | +2.8         |
>
> From the results and visualizations, we can draw the following conclusions:
>
> * We observe that gaze-guided prompting using gaze estimation leads to performance degradation in spatial questions. This suggests that spatial reasoning is highly sensitive to gaze estimation accuracy, as it often depends on precise spatial grounding. For instance, if the estimated gaze falls slightly off the object being manipulated—especially when nearby objects are close together, such as mistaking a knife for a cutting board—the model may misinterpret the target of the action and produce an incorrect response.
> * In contrast, temporal questions rely more on in-context temporal cues and are less affected by gaze estimation noise. Still, if the estimated gaze lands on a static background object rather than a newly appearing item, the model may miss key temporal transitions.
> * For causal reasoning, if the gaze is misestimated to a nearby object rather than the causal trigger (e.g., glancing at a mug instead of the hand turning on a kettle), the model might fail to infer the correct cause-effect relationship.
>
> We will add these insights to our analysis section to further clarify the differential impact of gaze quality across reasoning types. We hope our response sufficiently addresses your comments. If you have any further questions or concerns, please feel free to reach out to us for further discussion.
>
> **Reference**
>
> [1] Hansen, Dan Witzner, and Arthur EC Pece. "Eye tracking in the wild." Computer Vision and Image Understanding 98.1 (2005): 155-181.
>
> [2] Benita Wong, Joya Chen, You Wu, Stan Weixian Lei,Dongxing Mao, Difei Gao, and Mike Zheng Shou. Assistq:Affordance-centric question-driven task completion for egocentric assistant. In ECCV, pages 485–501. Springer, 2022.1, 4
>
> [3] Karttikeya Mangalam, Raiymbek Akshulakov, and Jitendra Malik. Egoschema: A diagnostic benchmark for very long-form video language understanding. Advances in Neural Information Processing Systems, 36, 2024.
>
> [4] Zhou, Sheng, et al. "Egotextvqa: Towards egocentric scene-text aware video question answering." Proceedings of the Computer Vision and Pattern Recognition Conference. 2025.
>
> [5] Zhou Yu, Dejing Xu, Jun Yu, Ting Yu, Zhou Zhao, Yueting Zhuang, and Dacheng Tao. Activitynet-qa:
> A dataset for understanding complex web videos via question answering. In Proceedings of the AAAI
> Conference on Artificial Intelligence, pages 9127–9134, 2019.
>
> [6] Fang, Xinyu, et al. "Mmbench-video: A long-form multi-shot benchmark for holistic video understanding." Advances in Neural Information Processing Systems 37 (2024): 89098-89124.

---

> > ### Comment · Reviewer_p3sj · 2025-08-05
> >
> > Thanks for the authors' response. I will raise to "borderline accept" rating.

---

> > ### Author Response · Authors · 2025-08-05
> >
> > Dear Reviewer p3sj,
> >
> > Thank you for taking the time to review our response. We sincerely appreciate your recognition of our work!
> >
> > Best regards,
> >
> > The Authors

---

### Official Review · Reviewer_GVAf · 2025-07-03

**Rating:** 5
**Confidence:** 4

**Summary:**

This paper introduces EgoGazeVQA, a novel benchmark for egocentric gaze-guided video question answering. It aims to evaluate whether MLLMs can leverage gaze information to better interpret user intention from first-person videos. The benchmark is constructed from three popular egocentric video datasets, augmented with gaze data and automatically generated QA pairs verified by human annotators. The authors further explore three gaze-guided prompting strategies and evaluate several open- and closed-source MLLMs.

**Dataset Code Accessibility:**

Yes

**Ethical Considerations:**

No, there are no or only very minor ethics concerns

**Final Justification:**

I will keep my "accept" rating.

**Limitations Weaknesses:**

1. There is no mention of how human performance compares to MLLMs on this benchmark. This makes it hard to judge whether the task is inherently ambiguous or model-limited.
2. Besides LoRA fine-tuning, could the authors also provide a comparison with full fine-tuning results?
3. We encourage the authors to provide a more detailed analysis of how gaze estimation errors impact model performance in specific spatial and temporal QA scenarios, which would offer deeper insights into the sensitivity of different reasoning tasks to gaze accuracy.

**Strengths Contributions:**

1. Overall this paper is well-written and is easy to understand.
2 This paper introduce EgoGazeVQA, an egocentric gaze-guided VQA benchmark that leverages gaze information to improve the understanding of longer daily-life videos.
3. Experimental results demonstrate that  the effectiveness of gaze-guided intent prompting methods in EgoGazeVQA.

---

> ### Author Rebuttal · Authors · 2025-07-31
>
> We thank the reviewer GVAf for the encouraging feedback and several points that can strengthen this paper. Below we provide the additional experiments analyses and results.
>
> **Q1: Human performance on this benchmark.**
>
> We thank the reviewer for the suggestion regarding human performance evaluation. To address this, we selected a representative subset from the test set and asked human volunteers to answer the QA pairs.
> To ensure a fair comparison and provide humans with access to the same information as the models, we visualized the gaze points directly on the video frames. The results are shown below:
>
> | Methods                    | Spatial | Temporal | Causal | Avg  |
> | -------------------------- |:-------:| :--------: | :------: | :----: |
> | Human                      | 71.7    | 67.6     | 91.2   | 76.8|
> | Qwen2.5VL-72B (baseline)   | 47.6    | 29.4     | 82.7   | 53.2 |
> | Qwen2.5VL-72B + GazeVisual | 50.0    | 35.3     | 85.3   | 56.8 |
>
> Interestingly, our benchmark also presents challenges to human subjects; however, humans consistently outperform the best-performing model (Qwen2.5-VL-72B) across all categories, particularly in spatial and temporal reasoning. This indicates that the task is limited by current model capabilities. The sizable gap indicates the opportunity for future work to better leverage gaze cues on intention understanding. We will add this results to our main paper.
>
> **Q2: Full fine-tuning results compared with LoRA fine-tuning.**
>
> We conducted full fine-tuning on Qwen2.5VL‑7B using both the EGTEA split and the Ego4D + EgoExo datasets. The results are presented below:
>
> | Methods       |Training Data| Test Data      | Spatial | Temporal | Causal | Avg  |
> | ------------- |-------------| -------------- | :-------: | :--------: | :------: | :----: |
> | Zero-shot     |N/A          | Ego4D + EgoExo | 40.4    | 43.3     | 78.2   | 54.0 |
> | LoRA SFT      |EGTEA        | Ego4D + EgoExo | 67.7    | 56.5     | **84.4**   | 69.5 |
> | Full SFT      |EGTEA        | Ego4D + EgoExo | **69.6**    | **58.9**     | 82.7   | **70.4** |
> | Zero-shot     |N/A           | EGTEA          | 45.7    | 32.9     | 77.3   | 52.0 |
> | LoRA SFT      |Ego4D + EgoExo| EGTEA          | 66.5    | 47.0     | 82.8   | 65.4 |
> | Full SFT      |Ego4D + EgoExo| EGTEA          | 69.5    | 48.2     | 77.9   | 65.2 |
>
> As shown, the performance improvement from full fine-tuning is comparable to LoRA, while requiring significantly more trainable parameters. This indicates parameter-efficient LoRA is still a competitive and practical choice.
>
>
> **Q3: Analysis of how gaze estimation errors impact model performance in specific spatial and temporal QA scenarios.**
>
> We appreciate the reviewer's suggestion to analyze the impact of gaze estimation errors. We break down the results by reasoning type: spatial, temporal, and causal. As shown in the table below (GT refers to using gaze ground truth in our prompting strategy, while GE refers to using gaze estimation.):
>
> | Category | w/o(baseline)  | GT   | $\Delta$(GT - w/o) | GE   | $\Delta$(GE - w/o) |
> | -------- | :----: | :----: | :------------: | :----: | :------------: |
> | Spatial  | 57.1 | 64.3 | +7.2         | 53.8 | -3.3         |
> | Temporal | 45.2 | 50.3 | +5.1         | 48.6 | +3.4         |
> | Causal   | 79.4 | 84.3 | +4.9         | 82.2 | +2.8         |
>
> From the results and visualizations, we can draw the following conclusions:
>
> * We observe that gaze-guided prompting using gaze estimation leads to performance degradation in spatial questions. This suggests that spatial reasoning is highly sensitive to gaze estimation accuracy, as it often depends on precise spatial grounding. For instance, if the estimated gaze falls slightly off the object being manipulated—especially when nearby objects are close together, such as mistaking a knife for a cutting board—the model may misinterpret the target of the action and produce an incorrect response.
> * In contrast, temporal questions rely more on in-context temporal cues and are less affected by gaze estimation noise. Still, if the estimated gaze lands on a static background object rather than a newly appearing item, the model may miss key temporal transitions.
> * For causal reasoning, if the gaze is misestimated to a nearby object rather than the causal trigger (e.g., glancing at a mug instead of the hand turning on a kettle), the model might fail to infer the correct cause-effect relationship.
>
> We will add these insights to our analysis section to further clarify the differential impact of gaze quality across reasoning types. We hope our response sufficiently addresses your comments. If you have any further questions or concerns, please feel free to reach out to us for further discussion.

---

> > ### Comment · Reviewer_GVAf · 2025-08-04
> > **Official Comment by Reviewer GVAf**
> >
> > Thanks for the authors' response. My concerns have been addressed. I will keep my "accept" rating.

---

> > ### Author Response · Authors · 2025-08-05
> >
> > Dear Reviewer GVAf,
> >
> > Thank you for taking the time to review our response. We sincerely appreciate your recognition of our work and your continued support.
> >
> > Best regards,
> >
> > The Authors

---

### Official Review · Reviewer_iBiG · 2025-07-03

**Rating:** 5
**Confidence:** 5

**Summary:**

This paper innovatively introduces gaze points into the first-person perspective VQA process, designing both a training and a test dataset. It also explores three gaze-guided prompting strategies and validates them across various scenes and tasks, yielding some very interesting results.

**Dataset Code Accessibility:**

Partly

**Dataset Code Comments:**

Not all testset and coordinate of gaze are relesed in that huggingface url.

**Ethical Considerations:**

No, there are no or only very minor ethics concerns

**Final Justification:**

I think the additional experiments and explanations have alleviated some of my concerns. Since this paper only proposes a training-free method for injecting gaze information through prompting, the supplementary experiments on “Textual Gaze Prompt + Question” also somehow demonstrate that this prompt is meaningful. I am inclined to give an accept score.

**Limitations Weaknesses:**

1. In Table 7, the baseline is not marked, and results without gaze estimation are missing, which would be helpful for comparison.

2. There is no analysis of out-of-distribution (OOD) performance. For example, how would the model perform on general tasks after training on gaze-related QA? All the QA tasks in this paper are gaze-related, but in real-world usage of multimodal large models, this is not the problem distribution. I strongly recommend including gaze-independent questions in the test set.

3. I think the red and green arrows in Tables 4, 5, etc., are not very meaningful. All your QA questions are about things like "where did you look?" Without gaze information, the model can only guess randomly. When the number of targets is large, the probability of a correct guess is lower, and when the number of targets is small, the probability of a correct guess is higher. Therefore, the distribution of these tasks and scenes might just reflect the distribution of the number of targets, which may not have significant analytical value.

**Strengths Contributions:**

This is the first paper to incorporate gaze into first-person VQA, making it a highly novel task that could potentially serve as an entry point for future human-computer interactions. The paper provides thorough validation across various models, including attempts to predict the gaze point, which led to notable performance improvements.

---

> ### Author Rebuttal · Authors · 2025-07-31
>
> We sincerely thank the Reviewer iBiG for the positive assessment and constructive suggestions. The feedback we received has been instrumental in further improving the quality of our paper. Below, we provide additional experimental results and explanations in response to your comments.
>
>
> **Q1: The baseline is not marked in Table 7.**
>
> We thank the reviewer for the helpful suggestion and we have updated the results of Qwen2.5VL‑72B performance across datasets in the following revised table below. GT refers to using gaze ground truth in our prompting strategy, while GE refers to using gaze estimation.
>
> | Datasets |  w/o(baseline) | GT | $\Delta$ (GT-w/o) |  GE | $\Delta$(GE-w/o) | MSE   |
> |---------|:----:|:--:|:-----------------:|:---:|:----------------:|:-----:|
> | Ego4D   | 60.1 | 67.0  | +6.9          | 60.5  | +0.4          | 0.154 |
> | EgoExo  | 61.7 | 67.3  | +5.6          | 64.5  | +2.8          | 0.038 |
> | EGTEA   | 59.3 | 64.0  | +4.7          | 58.7  | -0.6          | 0.119 |
> | Avg.    | 60.5 | 66.3  | +5.8          | 61.5  | +1.0          | 0.099 |
>
> This updated table does more clearly support the conclusion presented in the main paper: gaze estimation accuracy is strongly correlated with performance improvement. For example, in EgoExo (MSE = 0.038), both GT and GE settings yield notable gains. In contrast, when gaze estimation is less accurate, such as in EGTEA (MSE = 0.119), the benefit from GE diminishes drastically and may even slightly hurt performance. These results suggest that our proposed method is capable of effectively leveraging gaze signals—when they are reliable—to improve question answering.
>
> **Q2: Analysis of out-of-distribution (OOD) performance.**
>
> We appreciate the reviewer’s suggestion to evaluate out‑of‑distribution (OOD) performance. We do want to emphasize that most of the results in our main experiments are based on gaze-guided prompting without any parameter tuning. Since the model weights are frozen throughout our model, the generalization on other tasks or benchmarks won't be affected by our method.
>
> We would also like to clarify that the **generalization ability of MLLMs is typically assessed across multiple benchmarks, rather than expecting one single benchmark to span all domains and mimic real-world applications**. For instance, the MMMU benchmark focuses exclusively on math-related questions, and whether a model overfits to math tasks should be evaluated by its performance on other benchmarks. Similarly, in our context, there are several existing gaze-independent egocentric benchmarks that can be used to assess model generalization.
>
> Specifically, we conducted an additional experiment to evaluate the effect of LoRA fine-tuning on gaze-independent QA. We evaluated a LoRA-tuned Qwen2.5-VL-7B model on EgoSchema, which does not contain gaze annotations and thus does not support our gaze-guided prompting strategy. The baseline Qwen2.5-VL-7B model achieves an accuracy of 60.72\%, whereas LoRA fine-tuning on EGTEA drops the performance to 48.30\%, and fine-tuning on Ego4D and EgoExo further reduces it to 47.70\%. Notably, the poor generalization ability of the small model, combined with the limited dataset size, leads to overfitting and reduced performance on general egocentric VQA tasks. **We will include this discussion in the main paper and encourage future users to report results on additional related benchmarks to avoid overfitting solely to our benchmark**.
>
>
>
>
> **Q3: The complexity of the task purely based on number of targets in the scene.**
>
> We thank the reviewer for insightful comments on Tables 4 and 5. To address this point, we ran YOLOv11 on the entire test set to estimate how many discernible targets each clip contains. The average statistics are summarized below:
>
> | By Location      | Kitchen | Living Room | Medical Area | Garage | Others |
> | ---------------- | :------:| :---------: | :----------: | :----: | :----: |
> | objects classes  | 4.1     | 5.1         | 3.9          | 2.9    | 4.9    |
> | num objects      | 7.1     | 8.3         | 7.7          | 5.2    | 7.7    |
> | Qwen2.5VL-72B (acc)   | 58.5     | 60.9        | 70.8         | 69.4    | 59.9    |
>
> Taking Qwen2.5VL-72B as an example, we observe that although the *Living Room* contains more objects than the *Kitchen*, model performance is similar. Likewise, the *Medical Area* has a comparable number of objects to the *Kitchen*, yet achieves significantly higher accuracy.
>
> | By Activity      | Cooking | Gaming | Health | Mechanics | Creative | Others |
> | ---------------- | :-----: | :----: | :----: | :-------: | :------: | :----: |
> | objects classes  | 4.1     | 5.4    | 3.8    | 2.8       | 3.5      | 5.1    |
> | num objects      | 7.1     | 8.7    | 7.4    | 5.2       | 4.8      | 8.4    |
> | Qwen2.5VL-72B (acc)   | 58.7     | 65.3        | 71.2    | 69.4    | 52.3    |  52.1 |
>
> In activity-based scenes, the *Gaming* contains the most objects on average, but the model performs better than in *Cooking*, which has fewer targets.
>
> These findings suggest that **benchmark complexity is not solely determined by the number of visual targets. Instead, it is also shaped by the location and activity context**,  where different spatial structure, temporal dependencies, and the implicit goals of the human subject all affect how model interprets human intention.
>
> We hope this response adequately addresses your comments. We will include the above statistics as part of the released metadata to help users better understand the challenges posed by our benchmark. If you have any further questions or concerns, we would be happy to engage in additional discussion.

---

> > ### Comment · Reviewer_iBiG · 2025-08-07
> >
> > My main concern has not been fully addressed. It relates to the model’s ability to generalize to **general tasks**. According to the authors’ conclusion, introducing gaze currently unavoidably impacts the original model’s performance, causing it to lose a substantial portion of its fundamental capabilities while merely adding an interface for gaze-based user interaction.
> >
> > Secondly, the point raised in *Weaknesses 3* is mainly that models without gaze information are not suitable for answering your benchmark: asking them about gaze targets is pointless because they cannot accept such input.

---

> > > ### Author Response · Authors · 2025-08-08
> > > **Additional Clarifications and Generalization Experiments**
> > >
> > > We appreciate the reviewer iBiG for providing important follow up questions. We have several misunderstanding regarding the reviewer’s previous comments, and we sincerely apologize for this. We would like to provide additional responses to address the concerns.
> > >
> > > **On the claim that introducing gaze “unavoidably impacts” fundamental capabilities**
> > >
> > > We initially thought the Limitation2 was more about adding gaze-independent questions to our benchmark. We want to make two clarifications:
> > > - **Gaze-Guided Prompting approach (main results)**: Our main method is the intent-guided prompting, where the model weights are *frozen* and gaze is incorporated purely via prompting. In this training-free setting, the original model’s general capabilities remain *unchanged*, while it gains the ability to answer gaze-related queries.
> > >
> > > - **Lora Fine-tuning experiments (analytic purpose)**: The LoRA fine-tuning is intended as an **analytic experiment** — to study potential overfitting and to encourage researchers to explore more effective training methods for addressing EgoGazeVQA without overfitting — rather than as the optimal solution to our task. In fact, how to design a training pipeline that boosts performance on EgoGazeVQA without losing generalization ability on other tasks is an important open challenge we aim to introduce to the community for future exploration. We will clarify this in the revision.
> > >
> > > - **Additonal Generalization experiments**: Though solving the generalization problem is beyond the scope of this benchmark paper, we provide an initial attempt to optimize the LoRA SFT training data to better balance performance on EgoGazeVQA and EgoSchema. Instead of training solely on video–question–answer pairs, which risks overfitting, we concatenated the text-based gaze-guided prompt (GazeT) with each question, enabling the model to learn meaningful knowledge for gaze-specific associations while reducing overfitting on general questions.
> > >
> > > We applied this to Qwen2.5-VL-7B, and tested the model on EgoSchema for OOD generalization. Note that all LoRA SFT models are trained on EGTEA tested on Ego4D and EgoExo. The results are shown below:
> > >
> > > | Methods       |SFT Data Question Format|  EgoSchema Avg(\%)  | EgoGazeVQA Avg(\%) |
> > > | ------------- |---------------|-------------| ------------|
> > > | Zero-shot 7B     |N/A            |  60.72 | 50.26|
> > > | LoRA SFT  7B     |Question only  |  48.30 (-12.42) |68.85 (+18.59) |
> > > | LoRA SFT  7B     |Textual Gaze Prompt + Question|  58.32 (-2.40)|67.00 (+16.74) |
> > >
> > >
> > > For fair comparison, all above experiments from the above table are using textual gaze prompt (GazeV) for EgoGazeVQA. From the 7B model results, SFT using gaze-guided prompts together with questions helps mitigate overfitting compared with question-only training, achieving performance close to the original model on EgoSchema while also delivering notable improvements on EgoGazeVQA. Here, we focus on examining this data optimization strategy with textual gaze prompts (GazeT), and we will explore applying similar strategies to visual gaze prompt (GazeV) and sailency gaze prompt (GazeS) in future work.
> > >
> > >
> > > **On models without gaze input being pointless for this benchmark**
> > >
> > > We initially thought the comment from Limitation 3 referred to the concern that performance trends in Tables 4 and 5 might mainly reflect the number of possible targets rather than true task complexity. We apologize for this misunderstanding. Our benchmark is intentionally designed to evaluate a specific capability: using gaze information to answer egocentric intent-related questions. Including results for models without gaze serves a **baseline function** — it quantifies the performance gap between models that can exploit gaze and those that cannot. These baselines help calibrate expectations and guide future development, similar to the reviewer’s helpful suggestion on Table 7 to add baseline results without gaze estimation to calibrate performance changes. We will make the design intent of this experiment clearer in the paper to avoid misinterpretation.
> > >
> > > **Summary:**
> > > 1. Gaze-guided intent prompting is a training-free method and thus does not inherently degrade general capabilities.
> > > 2. We provide an optimized training data format that balances the performance on both EgoGazeVQA and EgoSchema.
> > > 3. The benchmark is intentionally scoped for gaze-aware MLLMs, with non-gaze baselines included solely to calibrate the necessity of incorporating gaze.

---

### Note · Authors · 2025-08-13

We thank the Area Chair (AC) for overseeing the review process and the reviewers for their constructive feedback, engagement during the discussion, and recognition of our work. After the rebuttal, we have  sufficiently responsed most of the reviewers’ concerns: Reviewer GVAf decided to maintain their “accept” rating, Reviewers p3sj and j14H raise their ratings to “borderline accept”.

**Response to Remaining Concerns in Latest Comments**:
Reviewer iBiG’s latest response still includes some remaining concerns. Their initial rating was “borderline accept”. We note that  their most recent comments differ substantially from their initial feedback: the initial comments focused on dataset distribution and complexity, which we responsed in our earlier rebuttal, whereas the follow-up comments focus on model performance and generalization ability. We provided detailed responses and additional experiments on these points in our latest comments, though the reviewer has not yet had the opportunity to review them due to timing.

We will incorporate all reviewers’ suggestions and clarifications into the final version of our paper. We sincerely thank the reviewers for their valuable contributions to improving our work.

Sincerely,

The Authors of Submission #91

---

### Decision · Program_Chairs · 2025-09-18

**Decision:**

Accept (poster)

**Comment:**

All reviewers incline to accept the submission thanks to the novelty, clear presentation and originality of the work. The AC agrees to the reviewers and recommends to accept the submission.